# Free Process Rewards without Process Labels

Lifan Yuan [*1]   Wendi Li [*2 3]   Huayu Chen [2]   Ganqu Cui [4]   Ning Ding [2]   Kaiyan Zhang [2]   Bowen Zhou [2]
Zhiyuan Liu [2]   Hao Peng [1]

## Abstract

Different from its counterpart outcome reward models (ORMs), which evaluate the entire responses, a process reward model (PRM) scores a reasoning trajectory step by step, providing denser and more fine-grained rewards. However, training a PRM requires labels annotated at every intermediate step, presenting significant challenges for both manual and automatic data collection. This paper aims to address this challenge. Both theoretically and empirically, we show that an *Implicit PRM* can be obtained *at no additional cost*, by simply training an ORM on the cheaper *response-level labels*. The only assumption is to parameterize the outcome reward as the log-likelihood ratios of the policy and reference models $r_\phi(\mathbf{y}) = \beta \log \frac{\pi_\phi(\mathbf{y})}{\pi_{\text{ref}}(\mathbf{y})}$, which can be optimized regardless of the specific choice of loss objectives. In experiments, we train our Implicit PRMs with various objectives and evaluate their performance on MATH. Implicit PRMs outperform strong MCTS-based baselines *á la* Math-Shepherd (Wang et al., 2023) using less than $1/38$ of the training data. We further find that scaling up instructions and responses benefits our Implicit PRMs, and the latter brings a larger gain. Particularly, Implicit PRMs with the cross-entropy (CE) loss is more data-efficient, and yields meaningful improvements even trained with only one response per instruction, a setup that suffers from extreme data scarcity and imbalance. We hope that our work will encourage a rethinking of PRM training approaches and contribute to making training PRMs more accessible.

*Equal contribution   [1]University of Illinois Urbana-Champaign  [2]Tsinghua University  [3]Huazhong University of Science and Technology  [4]Shanghai AI Lab. Correspondence to: Lifan Yuan <lifan4@illinois.edu>, Wendi Li <wendili@hust.edu.cn>, Ganqu Cui <cuiganqu@pjlab.org.cn>, Ning Ding <dn97@mail.tsinghua.edu.cn>.

*Proceedings of the $42^{nd}$ International Conference on Machine Learning*, Vancouver, Canada. PMLR 267, 2025. Copyright 2025 by the author(s).

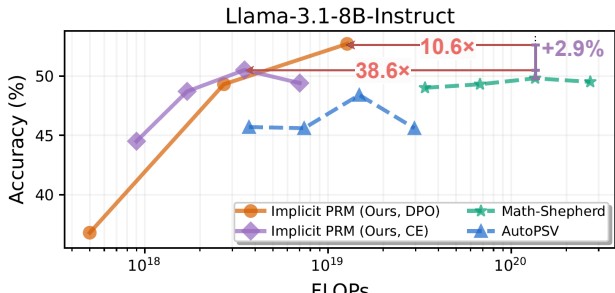

Figure 1: The x-axis indicates the FLOPs required to collect the data and train the model, and y axis the accuracies of best-of-64 performance. The accuracy is averaged over the best-of-64 accuracies of Mistral-7B-Instruct-v0.2 (Jiang et al., 2023), Llama-3.1-8B-Instruct, and Llama-3.1-70B-Instruct (Meta, 2024) on MATH (Hendrycks et al., 2021). Different dots on the same line indicates models trained with the same approach but on different scales of data. The top-left zone is desirable in this figure, as it suggests a model can achieve higher performance with less development overhead. Our implicit PRM is much cheaper to train while presenting the best performance under the same budget.

## 1. Introduction

Training on high-quality supervised data has driven the advances in LLMs development (Meta, 2024; Ding et al., 2023; Luo et al., 2024b; Yue et al., 2024; Yuan et al., 2024; Zhang et al., 2024b). Building upon this progress, reward models push the boundaries even further, especially in tasks requiring complex reasoning (Lightman et al., 2023; Wang et al., 2023; Snell et al., 2024). Outcome Reward Models (ORMs), designed to evaluate full responses, have been primarily explored, which can be used in both reinforcement learning (RL) and inference. However, due to the sparsity of outcome rewards, ORMs often yield suboptimal performance when reranking responses at inference (Lightman et al., 2023) and struggle with stability and efficiency during RL training (Cao et al., 2024; Chan et al., 2024). This highlights the growing demand for *denser and more fine-grained rewards*. Process Reward Models (PRMs), evaluating intermediate steps to provide fine-grained guidance, naturally meet this need. Existing work has shown consistent results

that PRMs outperform ORMs in best-of-N sampling (Wang et al., 2023; Snell et al., 2024) and RL (Setlur et al., 2024), and argues that scoring every intermediate step provides better transparency and interpretability (Leike, 2024).

Despite their promise, PRMs are much harder to train than ORMs, since collecting PRM training data requires annotating every intermediate step. To reduce human efforts, automatic annotation approaches have been proposed, where an intermediate step is labeled based on its estimated probability of leading to a correct outcome. Typically, this is achieved through either sampling massive look-ahead trajectories to estimate or directly training a verifier to predict Q value, both incurring extensive overhead (Wang et al., 2023; Lu et al., 2024). For example, collecting step-level data through sampling look-ahead trajectories as Wang et al. (2023) requires $38.8\times$ more FLOPs than training an ORM (§4).

We argue, from both theoretical and empirical perspectives, that building PRMs can be substantially cheaper than previously realized: **a strong PRM can be obtained at no additional cost from training an ORM on the cheaper response-level data, with a simple reward parameterization.** Specifically, by parameterizin the reward as the log-likelihood ratio of the policy and the reference models $r_\phi(\mathbf{y}) = \beta \log \frac{\pi_\phi(\mathbf{y})}{\pi_{\text{ref}}(\mathbf{y})}$, a common practice in DPO (Rafailov et al., 2023) and many of its variants (Azar et al., 2024; Ethayarajh et al., 2024; Chen et al., 2024; Rosset et al., 2024; Wu et al., 2024), a PRM can be automatically learned during ORM training. The process reward is then the same log-likelihood ratio, but calculated over a partial response. We dub our approach an **Implicit PRM** since it only requries response-level data and ORM training. Moreover, our insights are agnostic to the specific choice of the training objective, and are applicable to both DPO and all the variants that adopt the same form of implicit reward; it further extends to other objectives like the Cross-Entropy (CE) loss. This fresh theoretical insight generalizes the conclusion from Rafailov et al. (2024) that DPO training enables the model to learn the Q function; practically, our approach is particularly well-suited for scenarios where pairwise data is hard to obtain and algorithms like CE loss remain equally applicable, as shown in §5.1.

In experiments, we train our Implicit PRMs on a dataset consisting of 33K math instructions and eight solutions for each, and evaluate them through the best-of-N sampling on MATH (Hendrycks et al., 2021). We explore variants of our Implicit PRMs trained with different training objectives, including DPO, KTO (Ethayarajh et al., 2024), NCA (Chen et al., 2024), and CE (§4.2). All produce strong PRMs, outperforming competitive baselines including our reimplementations of Math-Shepherd (Wang et al., 2023) and AutoPSV (Lu et al., 2024) and six off-the-shelf open ORMs and PRMs,

with substantially better trade-offs between accuracy and development overhead, as shown in Figure 1. Particularly, when integrated into weighted best-of-N, CE stands as the most effective (§**??**). This makes CE loss appealing in scenarios where pairwise data is hard to collect, since it can handle unpaired and imbalanced data, and is demonstrated to be less data-consuming than DPO in order for an Implicit PRM with decent performance. Further, we find out that our Implicit PRM benefits from increased training data, and the scale of responses is more impactful than that of instructions (§5.1). Surprisingly, training on step-level data brings no further improvements to our Implicit PRMs (§C.2). Finally, we observe that, at least for the models and tasks we consider, the reference model can be omitted without hurting the model's quality (§5.3.2). This makes our Implicit PRMs even more appealing, offering improved training efficiency and performance without added inference overhead.

Bypassing the need for step labels, implicit PRMs substantially lower the data collection and training overhead of building PRMs while delivering stronger performance than existing methods. We hope that our work will encourage a rethinking of PRM training approaches and contribute to making training PRMs more accessible.

## 2. ORMs vs. PRMs: Dilemma of Performance and Expense

**Background** ORMs assign sparse rewards $r_\phi(\mathbf{y})$ to the entire response, and no feedback is provided until the last token is generated. In contrast, a PRM assesses the quality of every intermediate step and can provide reward after completing each (Lightman et al., 2023). Given an instruction and an $n$-step response $\mathbf{y}$ with $y_t$ being the $t$-th step and $\mathbf{y}_{<t}$ being the first $t-1$ steps, a PRM receives the concatenation of the instruction and the first $t-1$ steps, and assigns a reward to the $t$-th: $r_\phi^t(\mathbf{y}_{<t}, y_t)$. The Q value $q_\phi^t(\mathbf{y}_{<t}, y_t)$ indicates the expectation of outcome reward $r_\phi$ conditioned on the observed response $\mathbf{y}_{<t}$ and current step $y_t$. Lightman et al. (2023) define the process reward as the correctness of each step, while Wang et al. (2023) directly consider Q values as process rewards. We follow Lu et al. (2024) and define process reward as advantages, namely the difference between Q values: $r_\phi^t := q_\phi^t - q_\phi^{t-1}$. The benefits of adopting advantages as process rewards have been discussed by concurrent work (Setlur et al., 2024).

**PRMs outperform ORMs in both training and inference** Both ORMs and PRMs can provide rewards to assess model outputs. The dense step-level rewards from PRMs lead to stable and effective RL training (Cao et al., 2024; Chan et al., 2024), and perform better on reranking responses, with better transparency and interpretability. Also, ORMs are trained on complete responses, but the value model ini-

tialized from it only receives incomplete responses during RL training. On the contrary, PRMs are intrinsically trained to provide dense rewards given partial responses, thus the resulting value models may mitigate out-of-distribution issues that ORMs encounter.

**Training PRMs is substantially more expensive than ORMs**    Despite its effectiveness, training PRMs is more difficult due to challenges in training data collection. To collect training data for PRMs, MCTS is commonly used for automatic step annotation (Wang et al., 2023; Luo et al., 2024a). However, it introduces substantial extra cost. For MCTS-based step label annotation, a policy model will sample $N$ trajectories based on the concatenation of an instruction $x$ and partial response up to step $t$, each leading to a final answer (Wang et al., 2023). E.g., assuming 10-step rollouts and 8 subsequent trajectories for each step as in Wang et al. (2023), a total of $10 \times 8 = 80$ trajectories need to be generated to get step labels for each instruction, which is 80 times more than ORMs. Therefore, the scaling of PRMs is largely limited. Besides the overhead of training data collection, this MCTS approach can lead to suboptimal performance due to the noisy annotation process, as we will show below and in the experiments.

**MCTS estimation is not precise either**    We denote the set of correctness of subsequent trajectories as $\{c_1, c_2, \ldots, c_N\}$, each element being 0 or 1. Thereafter, two alternative label estimation strategies are available: (1) **Hard Estimation,** where step $t$ will be labeled as 1 if any rollout is correct and 0 otherwise: $l_t = \max\{c_1, c_2, \ldots, c_N\}$. (2) **Soft Estimation,** where step $t$ is labeled as the proportion of correct answers among all rollouts, namely $l_t = \sum_{t=1}^{N} c_t / N$. We refer the ORM used to judge the correctness of rollouts as $\phi$, the PRM trained on data from hard estimation as $\phi_h$, and the PRM trained on soft estimation data as $\phi_s$. If $\phi_h$ and $\phi_s$ are perfectly fitted, namely training losses reduced to 0, we have

$$q_{\phi_h}^t(\mathbf{y}_{<t}, y_t) = \max_{\mathbf{y}|\mathbf{y}_{\leq t}} r_\phi(\mathbf{y}),$$
$$q_{\phi_s}^t(\mathbf{y}_{<t}, y_t) = \mathbb{E}_{\pi_{ref}(\mathbf{y}|\mathbf{y}_{\leq t})} r_\phi(\mathbf{y}) \tag{1}$$

However, both estimation strategies may be noisy. Specifically, $q_{\phi_h}^t$ represents the maximum outcome reward $r_\phi$ given $\mathbf{y}_{<t}$, rather than the expectation, thus overestimating the Q value; For $q_{\phi_s}^t$, given the limited capability of the policy model in practice, it can be challenging to sample correct solutions for difficult instructions, suffering from false negative noises and thus underestimating Q.

## 3. Implicit PRMs For Free Through Reward Parameterization

In this section, we show that an ORM can directly represent an expectation of the outcome reward by itself by simple

reward parameterization.    In other words, a PRM can be inherently derived from the same ORM without any dedicated training, offering better performance than MCTS-based approaches with substantially lower overhead.

**Reward parameterization in existing work**    Current literature typically parameterize rewards by either (1) the linear transformation of hidden states, with the reward model being a sequence classifier (Ouyang et al., 2022; Touvron et al., 2023; Zhu et al., 2023; Cui et al., 2024) or (2) generative logits, with reward models being an auto-regressive LM and trained to predict the label of partial or complete responses as "good" or "bad" tokens, and sometimes a third "neutral" (Zhang et al., 2024c; Mahan et al., 2024; Lightman et al., 2023; Wang et al., 2023; Luo et al., 2024a).

Unfortunately, under either of the two parameterizations, PRMs would require expensive step labels to train. To address this issue, **we propose to train an ORM with implicit reward modeling, which will automatically enable a PRM regardless of the loss functions.** Next, we illustrate this in detail:

**Proposition 3.1.** *(Proof in Appendix A) Consider an ORM where the reward is parameterized by the log-likelihood ratio of two causal LMs, i.e. $r_\phi(\mathbf{y}) := \beta \log \frac{\pi_\phi(\mathbf{y})}{\pi_{ref}(\mathbf{y})}$. Define $q_\phi^t(\mathbf{y}_{<t}, y_t) := \sum_{i=1}^{t} \beta \log \frac{\pi_\phi(y_i|\mathbf{y}_{<i})}{\pi_{ref}(y_i|\mathbf{y}_{<i})}$. $q_\phi^t$ is the exponential average of $r_\phi$ at step $t$.*

$$q_\phi^t(\mathbf{y}_{<t}, y_t) = \beta \log \mathbb{E}_{\pi_{ref}(\mathbf{y}|\mathbf{y}_{\leq t})} e^{\frac{1}{\beta} r_\phi(\mathbf{y})} \tag{2}$$

*Hence, $q_\phi^t$ represents an exact expectation of outcome reward $r_\phi$ at step $t$, i.e., the Q value.*

Proposition 3.1 indicates that when modeling $r_\phi(\mathbf{y}) := \beta \log \frac{\pi_\phi(\mathbf{y})}{\pi_{\mathrm{ref}}(\mathbf{y})}$ to train an ORM with the standard pipeline, where $\beta$ is a hyperparameter, $\phi$ can implicitly learn a Q function. Hence, process reward $r_\phi^t$ can be obtained by:

$$r_\phi^t := q_\phi^t - q_\phi^{t-1} = \beta \log \frac{\pi_\phi(y_t|\mathbf{y}_{<t})}{\pi_{\mathrm{ref}}(y_t|\mathbf{y}_{<t})} \tag{3}$$

Notably, this conclusion still holds when $y_t$ represents the $t$-th token rather than step $t$. **This gives us an inspiring hint: we can indeed obtain PRMs of any granularity simply by collecting response-level data and training an ORM, without any burden of annotating step labels, as shown in Figure 2.** The proposition is agnostic to specific choices of the training objective of ORMs. It can be instantiated with different objectives as vanilla ORM training, with the only difference being substituting the $r_\phi(\mathbf{y})$ with $\beta \log \frac{\pi_\phi(\mathbf{y})}{\pi_{\mathrm{ref}}(\mathbf{y})}$. Particularly, many existing preference learning algorithms have already met our assumption (Rafailov et al., 2023; Azar et al., 2024; Ethayarajh et al., 2024; Chen et al., 2024; Wu et al., 2024).

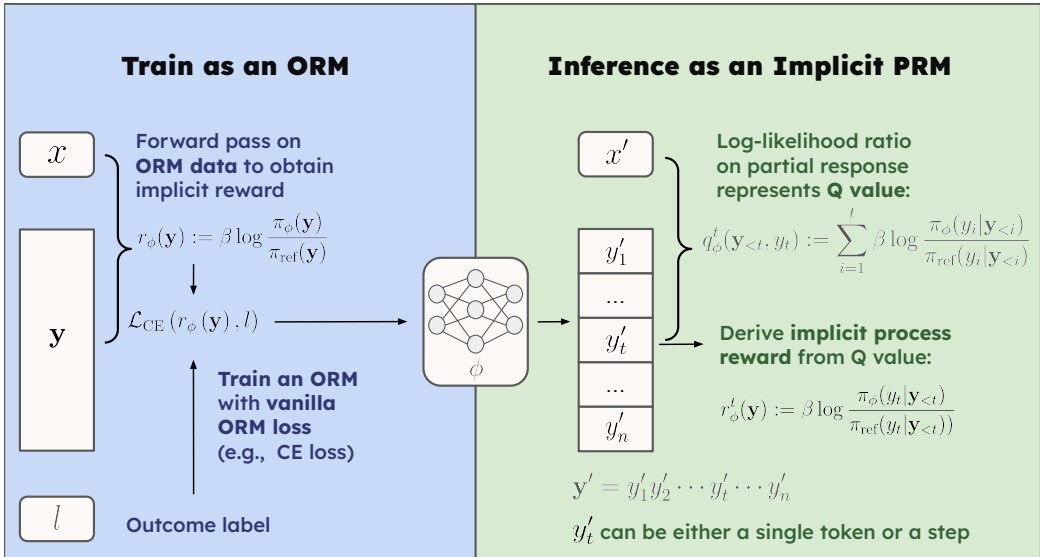

Figure 2: With specifically designed reward representation, we can train an ORM with a standard pipeline while automatically enabling an Implicit PRM.

Besides making PRM training more accessible, our implicit process reward can be more accurate than those derived from $q_{\phi_s}^t$ and $q_{\phi_h}^t$ in Eq. 1 (Wang et al., 2023), as indicated by the following proposition:

**Proposition 3.2.** *The performance of $q_\phi^t$ is guaranteed by the following conditions: $q_\phi^t$ is bounded by $q_{\phi_s}^t$ and $q_{\phi_h}^t$, and can reach these bounds with specific values of $\beta$. That is,*

$$q_{\phi_s}^t = \mathbb{E}_{\pi_{ref}(\mathbf{y}|\mathbf{y}_{\leq t})} r_\phi(\mathbf{y}) \leq q_\phi^t(\mathbf{y}_{<t}, y_t) \leq \max_{\mathbf{y}|\mathbf{y}_{\leq t}} r_\phi(\mathbf{y}) = q_{\phi_h}^t$$

(4)

*holds. The left-hand equality is attained as $\beta \to \infty$ and the right-hand one is attained as $\beta \to 0$.*

Proposition 3.2 demonstrates that $q_\phi^t$ ranges between the soft-estimated and hard-estimated Q values annotated by MCTS-based approaches. The above bounds suggest that our approach has better accuracy and robustness to noises than MCTS-based approaches. Specifically, as discussed in §2, $q_{\phi_h}^t$ overestimates the Q value while $q_{\phi_s}^t$ underestimates Q due to false negative noises. Since $q_\phi^t$ lies between $q_{\phi_h}^t$ and $q_{\phi_s}^t$, it could potentially mitigate both issues and estimate the Q value more accurately. Concurrent work defines our $q_\phi^t$ as an entropy regularized process reward and has empirically shown its superiority over $q_{\phi_s}^t$ and $q_{\phi_h}^t$ on best-of-N sampling (Zhang et al., 2024a).

**Connection to Rafailov et al. (2024)** An intuition similar to Proposition 3.1 has been brought up by Rafailov et al. (2024), which demonstrates that DPO enables models to learn the Q function implicitly, but our insights subsume their conclusion since this property is not limited to the DPO algorithm. For example, given response-level label $l$, we can further generalize to cross-entropy (CE) loss to handle

practical scenarios with unpaired and imbalanced data:

$$\mathcal{L}_{CE} = -l \cdot \log \sigma \left( \beta \log \frac{\pi_\phi(\mathbf{y})}{\pi_{\text{ref}}(\mathbf{y})} \right)$$
$$- (1 - l) \cdot \log \left[ 1 - \sigma \left( \beta \log \frac{\pi_\phi(\mathbf{y})}{\pi_{\text{ref}}(\mathbf{y})} \right) \right]$$

(5)

**Reference Model** One difference between our modeling of rewards and previous ones is the incorporation of a reference model $\pi_{\text{ref}}$. We acknowledge that this comes at an inference cost: to calculate the reward, both the policy and reference model are served, which doubles the inference cost than vanilla PRM. However, it is prevalent in existing preference learning algorithms and works as the KL constraint to prevent the policy model $\pi_\phi$ deviating too far from its starting checkpoint. Moreover, it is less a problem in practice, as we will show in §5.3.1 that a large proportion of the inference overhead in best-of-N sampling comes from the generation model, especially when the generation model is much larger than the reward model. Further, we also show in §5.3.2 that when the Implicit PRM is built from a strong model that has undergone preference learning, such as Llama-3.1-Instruct, excluding $\pi_{\text{ref}}$ leads to little or no accuracy drop. This makes our approach appealing in practice since it can achieve better accuracy than existing PRMs with exactly the same inference overhead, but substantially lower development overhead.

## 4. Experiments

### 4.1. Setup

**Evaluation** Following standard practice (Lightman et al., 2023), we evaluate PRMs with best-of-N (BoN) on MATH-

500 (Hendrycks et al., 2021). To study the generalizability of the PRMs, we test each PRM using three generation models with different levels of capabilities: Mistral-Instruct-v0.3 (Jiang et al., 2023), Llama-3.1-8B-Instruct, and Llama-3.1-70B-Instruct (Meta, 2024). For each completion, we apply PRMs to score each step and pick the lowest step reward as the score for overall responses. We also compare the development overhead of the models in terms of FLOPs, including those required in both the automatic data collection and PRM training.

**Training dataset** Unless stated otherwise, we adopt the following training setup throughout all experiments: We use math instructions from UltraInteract (Yuan et al., 2024) and sample eight rollouts per instruction using Llama-3.1-8B-Instruct, and then assess rollout correctness with ground truths. We train PRMs based on Llama-3.1-8B-Instruct with $\beta = 0.05$, which is empirically determined.

**Implicit PRM instantiation** As demonstrated in §3, our approach can be instantiated with any reward modeling objective with the reward parameterized as $r_\phi := \beta \log \frac{\pi_\phi(\mathbf{y})}{\pi_{\text{ref}}(\mathbf{y})}$. We explore various objectives that meet the requirements, including DPO (Rafailov et al., 2023), KTO (Ethayarajh et al., 2024), NCA (Chen et al., 2024), and the cross-entropy (CE) loss. Please refer to Eq. 5 for the implementation of CE loss. For DPO and NCA, we pair each correct rollout with an incorrect counterpart and train our RM on these response-level pairs, while for KTO and CE loss, we directly train on the unpaired and imbalanced rollouts, which is more general in practical scenarios. We also implement two data balanced setup for CE to analyze the impact of pairwise data, i.e. balancing the positive and negative responses simply for the entire dataset, or more strictly for the each each instruction. We denote the two setups as Dataset-wise Balanced and Instruction-wise Balanceed.

**Baselines** Our baselines include our implementation of existing methods and off-the-shelf open models. We reimplement Math-Shepherd (Wang et al., 2023) and AutoPSV (Lu et al., 2024) for fair comparisons, representative algorithms in their categories. Math-Shepherd annotates step labels using MCTS estimations as illustrated in §2. AutoPSV annotates steps with a two-stage strategy. It firsts trains an outcome supervision verifier (OSV) that predicts Q value for each step, then use the OSV to annotate step labels. A PRM is obtained by continual training on the OSV with process labels. We also compare to six off-the-shelf ORMs and PRMs, namely EurusRM-7B (Yuan et al., 2024), SkyworkRM-Llama3.1-8B (Liu et al., 2024), ArmoRM-Llama3-8B (Wang et al., 2024), Math-Shepherd-7B (the offical release of Wang et al., 2023), RLHFlow-8B-Mistral-

Data[1], and RLHFlow-8B-DS-Data[2]. We note that these off-the-shelf baselines are trained on different instructions and responses, while our two reimplementations are trained on the same data as our Implicit PRM.

## 4.2. Results

**Various implicit reward modeling objectives outperform baselines** According to BoN results shown in Table 1, all four variants of our Implicit PRMs consistently improve the accuracies of the three different generation models. Among them, DPO achieves an averaged accuracy of 50.4, performing better in general, closely followed by NCA with an averaged accuracy of 49.4. CE presents strong performance, despite that it is trained on unpaired and imbalanced data. Specifically, with an averaged accuracy of 48.4, it beats our implemented Math-Shepherd and AutoPSV by 0.6 and 2.7 respectively, and outperforms other open-source reward models except RLHFlow-8B-Mistral-Data and RLHFlow-8B-DS-Data, both of which achieves 49.1. This indicates the potential in empowering real-world applications where pairwise data is hard to collect. Nevertheless, according to CE versus CE (Inst.-wise Balanced), it is still beneficial to have balanced positive and negative responses for each instruction in the training dataset, which aligns with conventional understandings on CE as a classification loss. However, comparing CE (Dataset-wise Balanced) to CE, simply balancing the entire dataset by randomly filtering examples of the class with more data can be detrimental.

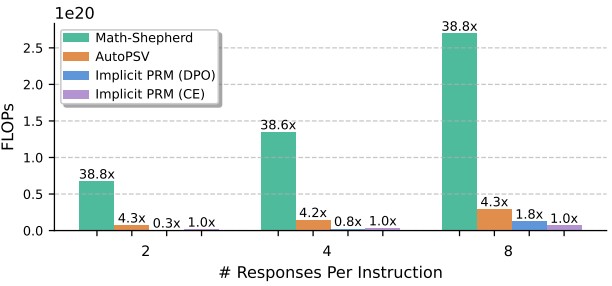

Figure 3: Overhead of developing different PRMs, in terms of FLOPs during data collection and training (lower is better). The x axis is the number of responses per instruction determining the scale of training data, and the y axis the FLOPs. Our implicit PRM always consumes the least FLOPs, with CE being $38.6\times$ to $38.8\times$ more efficient than Math-Shepherd across different dataset scales.

**Our Implicit PRMs reduce the overhead of data collection and training by** $38.8\times$ As shown in Figure 3, **with**

[1]https://huggingface.co/RLHFlow/Llama3.1-8B-PRM-Mistral-Data
[2]https://huggingface.co/RLHFlow/Llama3.1-8B-PRM-DeepSeek-Data

Table 1: Different reward models' best-of-N sampling performance on MATH test set with three different generation models. When completing instructions with a temperature of 0.5, the three generation models' accuracies are 9.6%, 44.6%, and 63.2% respectively.

| Type | Reward Model | Mistral-7B-Inst-v0.2 Pass@1: 9.6 | | | Llama-3.1-8B-Inst Pass@1: 44.6 | | | Llama-3.1-70B-Inst Pass@1: 63.2 | | | Avg. |
|------|--------------|------|------|------|------|------|------|------|------|------|------|
| | | @4 | @16 | @64 | @4 | @16 | @64 | @4 | @16 | @64 | |
| | | *Open-Source Reward Models* | | | | | | | | | |
| **ORM** | EurusRM-7B | 17.2 | 21.0 | 20.4 | 49.6 | 51.6 | 51.8 | 69.0 | 69.6 | 72.2 | 46.9 |
| | SkyworkRM-Llama3.1-8B | 16.0 | 19.6 | 23.4 | 49.0 | 50.4 | 48.2 | 70.4 | 72.6 | 72.0 | 46.8 |
| | ArmoRM-Llama3-8B | 16.6 | 21.0 | 23.2 | 47.8 | 48.6 | 49.4 | 70.6 | 70.8 | 71.0 | 46.6 |
| **PRM** | Math-Shepherd-7B | 16.0 | 21.0 | 20.4 | 50.0 | 52.4 | 52.8 | 66.4 | 65.8 | 65.6 | 45.6 |
| | RLHFlow-8B-Mistral-Data | **19.4** | **25.2** | **30.2** | 51.8 | 52.0 | 50.6 | 70.8 | 71.0 | 71.2 | 49.1 |
| | RLHFlow-8B-DS-Data | 17.2 | 23.0 | 25.2 | **54.4** | 54.2 | 55.8 | 68.6 | 70.4 | **73.0** | 49.1 |
| | | *Our Implementations* | | | | | | | | | |
| **Baselines** | **Math-Shepherd** | 17.6 | 24.4 | 26.8 | 50.0 | 51.4 | 52.8 | 68.6 | 69.4 | 68.8 | 47.8 |
| | **AutoPSV** | 16.6 | 20.6 | 22.2 | 52.2 | 51.4 | 52.2 | 68.4 | 65.4 | 62.4 | 45.7 |
| **Implicit PRM** | **DPO** | 18.6 | 24.4 | 28.8 | 54.0 | **55.4** | **57.0** | 71.8 | 71.2 | 72.2 | **50.4** |
| | **KTO** | 15.6 | 18.4 | 18.6 | 49.6 | 51.8 | 50.8 | **72.6** | 67.0 | 67.2 | 45.7 |
| | **NCA** | 18.6 | 23.8 | 28.0 | 52.4 | 53.4 | 55.2 | 69.0 | **73.0** | 71.6 | 49.4 |
| | **CE** | 18.8 | 24.0 | 28.0 | 52.6 | 54.4 | 53.0 | 70.6 | 67.0 | 67.2 | 48.4 |
| | **CE (Dataset-wise Balanced)** | 18.0 | 23.6 | 27.0 | 52.6 | 54.2 | 52.6 | 68.6 | 66.8 | 67.0 | 47.8 |
| | **CE (Inst.-wise Balanced)** | 17.6 | 22.6 | 26.2 | 52.6 | 55.2 | 54.6 | 69.4 | 71.2 | 72.0 | 49.0 |

three different training data scales. **Math-Shepherd generally costs 38.8x more FLOPs than the Implicit PRM (CE).** Compared to Implicit PRM (DPO), the number becomes 146.5x, 49.9x, and 21.3x under different number of responses per instruction respectively.

We plot the scaling trends of the average performance of each method with corresponding number of tokens consumed in Figure 1, from which we can clearly see that our Implicit PRMs achieve better performance with much less data collection and training overhead.

## 5. Analysis

### 5.1. Scaling Instructions and Responses Helps

**Setup** We conduct scaling analysis with DPO and CE on both instructions and responses of the training dataset. For instruction scaling, we randomly downsample 25%, 50%, and 75% instructions to train our Implicit PRM. For response scaling, since DPO can only train on paired responses, we train models with 2, 4, and 8 rollouts; while for CE, we also implement training with *only one rollout per instruction*, the extreme case of unpaired setup.

**Results** We present results in Figure 4 and Figure 5 respectively. Takeaways are summarized as follows: (1) **Scaling instructions and responses consistently**

**improve the performance of our Implicit PRM.** The trend is particularly clear on Mistral-7B-Inst-v0.2 and Llama-3.1-8B-Inst, but there are also a few outliers on Llama-3.1-70B-Inst. (2) **Compared to instructions, scaling up responses seems to be more influential on Implicit PRMs**, as reflected by the larger performance variations between the minimum and maximum data setups. Taking a closer look at the response scaling, (3) **DPO requires more data to obtain a decent performance than CE.** From Figure 5, DPO is under-trained with two responses per instruction, which can be partly attributed to the insufficient amount of instructions: two responses may not constitute a pair to train our DPO variant, and thus many instructions can not be used in training. In contrast, CE generally performs better with insufficient data and can always improve different generation model, even when it is trained with one response per instruction with pairs, the extreme case of the unpaired setup. This presents a huge advantage in real-world data scarcity scenarios.

### 5.2. PRM Ability Does Not Translate into Policy Performance

Implicit PRM is trained in an auto-regressive manner, sometimes directly using preference learning algorithms, which are primarily used to improve policy models. Therefore, it reserves the nature as a causal LM and can still serve as

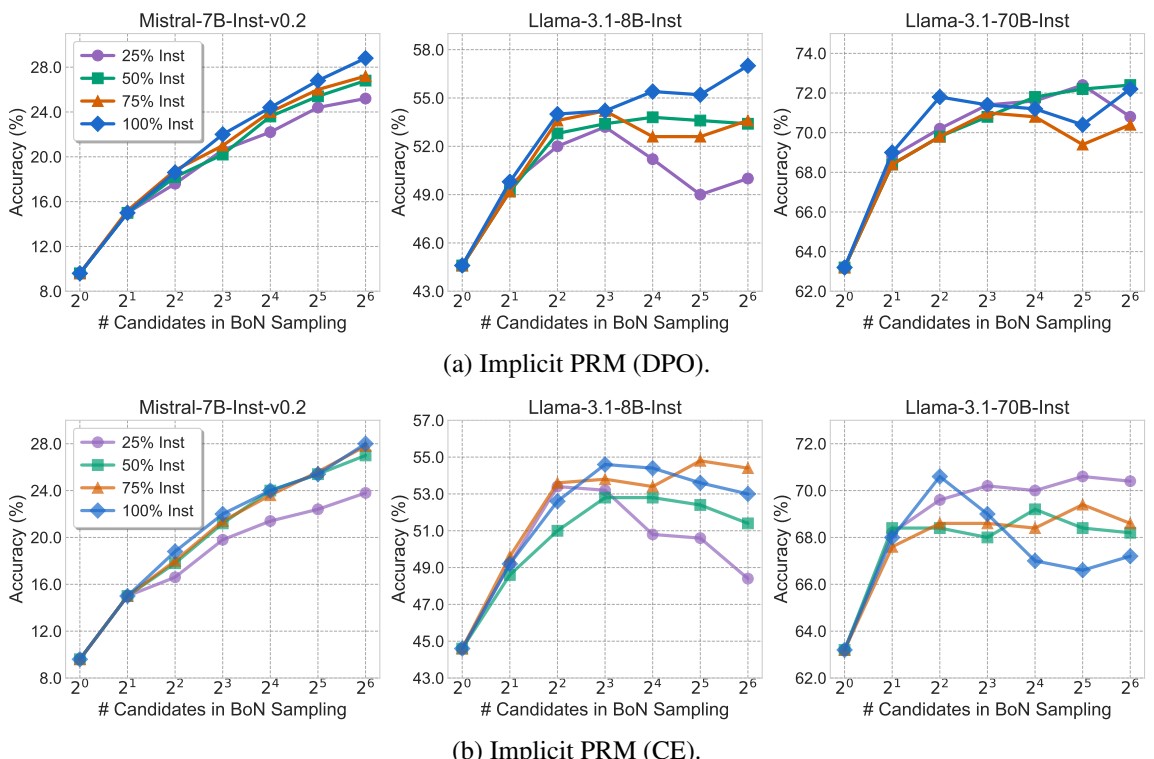

(a) Implicit PRM (DPO).

(b) Implicit PRM (CE).

Figure 4: Scaling instruction numbers. Our implicit PRM's performance on Mistral-7B-Instruct-v0.2 and Llama-3.1-8B-Instruct scales well with the number of instructions, despite the trend is more complex on Llama-3.1-70B-Instruct.

a policy model to solve downstream problems directly. In this section, we test on MATH500 (Hendrycks et al., 2021; Lightman et al., 2023) to analyze the correlation between their PRM ability and performance as a policy model.

According to Table 2, only trainiing with KTO leads to an improvement on MATH500, compared to Llama-3.1-8B-Instruct. Interestingly, based on Table 1, KTO performs the worst as an Implicit PRM. In contrast, DPO and CE, the two algorithms that per-

Table 2: Implicit PRMs' performance on MATH500 when used to solve the problems directly.

| Model | Accuracy |
|---|---|
| Llama-3.1-8B-Inst | 45.2 |
| + DPO | 25.8 |
| + KTO | 46.6 |
| + NCA | 35.6 |
| + CE | 28.6 |

form the best in without majority voting and with majority voting setups, respectively, achieve the lowest accuracies. This indicates that PRM ability does not improve as the policy model improves, and there can even be an unexpected trade-off between the both abilities.

### 5.3. Inference Overhead of the Reference Model

Our approach needs of an additional reference model in inference. However, we show that the the reference model does not double overall inference overhead in practice, es-

Table 3: GPU time costs during best-of-N sampling relative to the cost of generation model (%). The overall inference overhead of baselines on three test sets are 66.6%, 70.8%, and 90.9% of that of our implicit PRM, respectively. Namely, the reference model does not double the inference cost in practice, and the extra inference overhead becomes more marginal as the generation model gets larger.

| Source of Cost | Method | Mistral-7B-Inst-v0.2 | Llama-3.1-8B-Inst | Llama-3.1-70B-Inst |
|---|---|---|---|---|
| Generation Model | - | 100.0 | 100.0 | 100.0 |
| Reward Model | Baselines | 33.5 | 29.4 | 9.1 |
| | Implicit PRM | 201.6 | 141.7 | 22.2 |
| Total | Baselines | 200.9 | 171.1 | 111.1 |
| | Implicit PRM | 301.6 | 241.7 | 122.2 |

pecially when the generation model is much larger than the reward model (§5.3.1). Surprisingly, the reference model can be removed at inference in certain cases (§5.3.2).

#### 5.3.1. THE REFERENCE MODEL DOES NOT DOUBLE OVERALL INFERENCE OVERHEAD

**Setup** We calculate the time costs of best-of-N sampling on MATH500 in practice. The entire process includes (1) generating multiple candidate solutions to the instruction using the generation model, and (2) scoring each candidate using a PRM. We use vLLM (Kwon et al., 2023) to implement the former and Huggingface Accelerate (Gugger et al., 2022) for the latter.

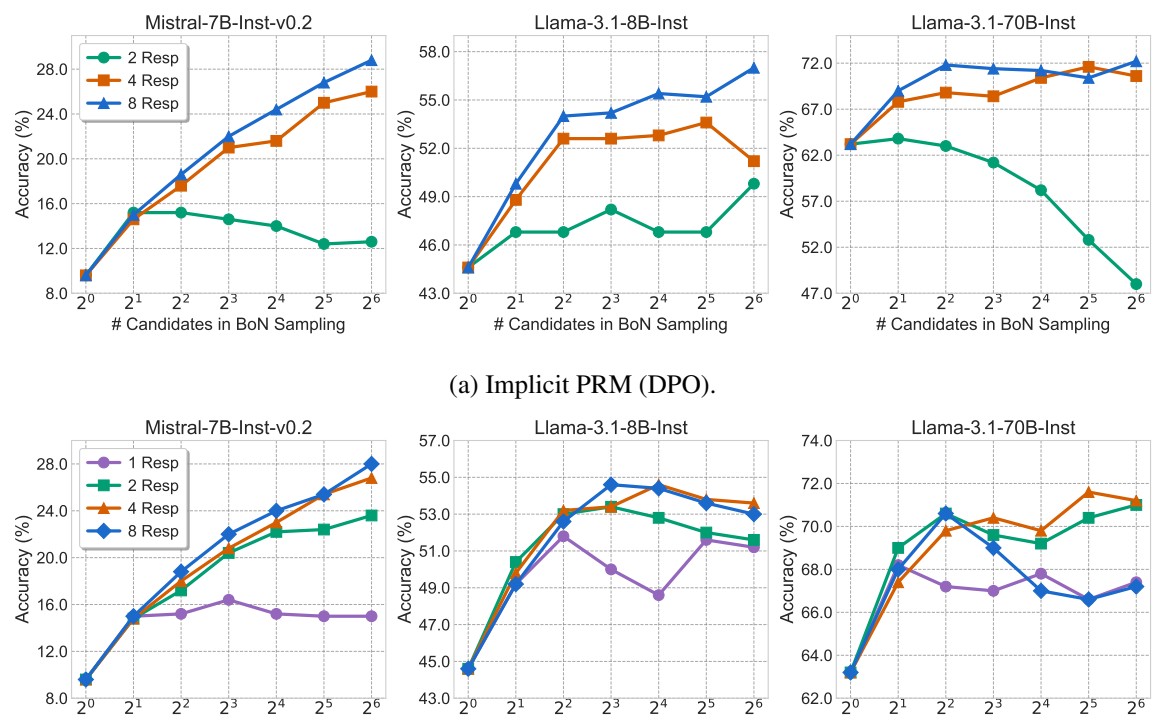

(a) Implicit PRM (DPO).

(b) Implicit PRM (CE). Note that one repsonse per instruction is the extreme case of the unpaired setup.

Figure 5: Scaling responses number for each instruction. Our implicit PRM generally benefits from scaling up the number of responese for each instruction. Particularly, DPO is under-trained with two responses per instruction. This can be partly attributed to the insufficient amount of instructions: two responses may not constitute a pair to train our DPO variant, and thus many instructions can not be used in training. In contrast, CE generally performs better with insufficient data and can always improve different generation model, even when it is trained with one response per instruction with pairs.

**Results** We present the GPU time costs on an A100 80G relative to that of the generation model in Table 3. We find that the inference overhead from generation model takes a large proportion of the total overhead, especially when the generation model is much larger than the reward model. Therefore, the overall inference overhead of baselines on three test sets are 66.6%, 70.8%, and 90.9% of that of ours, respectively. It is noteworthy that the extra overhead introduced by the reference model becomes more marginal as the generation model b larger, and is almost negligible when Llama-3.1-70B-Instruct serves as the generation model.

### 5.3.2. THE REFERENCE MODEL CAN BE REMOVED AT INFERENCE IN CERTAIN CASES

We note that our proposition still holds under a uniformly distributed reference model, i.e. $\log \pi_{\text{ref}} = constant$. In best-of-N sampling, only relative scores between steps or responses matter, where the constant $\log \pi_{\text{ref}}$ can be canceled out, equivalent to exclude the reference model in reward parametrization. Therefore, we derive a more

efficient implementation of our proposition by removing the reference model. We examine its effectiveness and explore if we can simply our method to reduce the inference overhead in practice.

**Setup** To this end, we explore two model training configurations: parameterizing the outcome reward either with or without a reference model. . We then apply both models to best-of-N sampling and evaluate whether including the reference model has any impact to the performance. We also compare to directly using Llama-3.1-8B-Instruct, the reference model in our Implicit PRM in previous experiments, as the reward model. It serves as a controlled baseline without any RM training on our data, but has undergone preference learning (Meta, 2024).

**Results** Surprisingly, no performance degradation is observed when the reference model is ablated in both training and inference, suggesting a more practically efficient variant of our approach. Besides, Llama-3.1-8B-Instruct achieves strong performance too. This potentially explains why the

Table 4: Ablating reference model in both training and inference. Neither consistently hurts our implicit PRM. More surprisingly, the reference model, Llama-3.1-8B-Instruct, already perfroms well on Best-of-N sampling.

| Setup | | Mistral-7B-Inst-v0.2 | | | Llama-3.1-8B-Inst | | | Llama-3.1-70B-Inst | | | Avg. |
| --- | --- | --- | --- | --- | --- | --- | --- | --- | --- | --- | --- |
| Train | Inference | @4 | @16 | @64 | @4 | @16 | @64 | @4 | @16 | @64 | |
| Llama-3.1-8B-Instruct | w/o Ref | 14.8 | 16.2 | 18.4 | 49.0 | 50.4 | 52.2 | 69.6 | 71.0 | 71.0 | 45.8 |
| + DPO w/ Ref | w/ Ref | 18.6 | 24.4 | 28.8 | 54.0 | 55.4 | 57.0 | 71.8 | 71.2 | 72.2 | 50.4 |
| | w/o Ref | 17.8 | 23.4 | 27.8 | 54.2 | 56.6 | 57.6 | 71.6 | 73.6 | 73.2 | 50.6 |
| + DPO w/o Ref | w/ Ref | 17.8 | 23.4 | 28.4 | 54.0 | 55.2 | 57.6 | 70.6 | 72.0 | 73.2 | 50.2 |
| | w/o Ref | 17.4 | 22.6 | 25.6 | 54.8 | 56.4 | 58.2 | 70.4 | 73.2 | 74.0 | 50.3 |

reference model can be removed: The reference model is already capable of appropriately assigning high rewards to "good" steps and low ones to "bad" steps. Recall the process reward is $\sum_{i=t-1}^{t} \beta \log \pi_\phi(y_i|\mathbf{y}_{<i})/\pi_{\text{ref}}(y_i|\mathbf{y}_{<i})$. Intuitively, a good step might receive high probabilities by both $\pi_\phi$ and $\pi_{\text{ref}}$, and therefore lowering its reward; on the other hand, a bad step might receive low probabilities by both, thereby increasing its reward. This creates confusion to the PRM. We argue that this behavior is actually beneficial during RL training: when the reference model $\pi_{\text{ref}}$ already performs well on certain actions, smaller rewards and consequently smaller policy gradients prevent over-training the policy model $\pi_\phi$ on these already-optimized actions. Nevertheless, it is undesired on such inference-time response selection tasks. This suggests that our Implicit PRM is particularly appealing in practice, since most of the time practitioners will build their PRMs from a strong reference model such as Llama-3.1-8B-Instruct. In such cases, $\pi_{\text{ref}}$ can be dropped in inference without hurting the performance as the above results suggest, and **our approach can achieve stronger performance than baselines with substantially cheaper training, without introducing any additional inference overhead.**

## 6. Related Work

**Complex Reasoning of LLMs**  Complex reasoning has become a key capability of Large Language Models (LLMs) yet remains challenging even to state-of-the-art ones (Jimenez et al., 2024; Tian et al., 2024). Various techniques have been explored to improve LLMs on reasoning throughout different stages of their lifecycles, such as pre-training (Azerbayev et al., 2024; Paster et al., 2024; Li et al., 2023), post-training (Luo et al., 2024b; Yue et al., 2024; Yuan et al., 2024; Meta, 2024; Ouyang et al., 2022), and inference (Wei et al., 2022; Fu et al., 2023; Hao et al., 2023; Lightman et al., 2023). Among them, the process reward model , has attracted recent attention for its effectiveness in a variety of settings (Lightman et al., 2023).

**Implicit Reward**  Implicit reward has already been widely adopted in preference learning. Most existing works focus on applying these algorithms to align models on top of supervised fine-tuning (Rafailov et al., 2023; Azar et al., 2024; Ethayarajh et al., 2024; Chen et al., 2024; Rosset et al., 2024; Wu et al., 2024); recent work also tries to leverage the implicit rewards of resulting models as outcome rewards (Lambert et al., 2024; Zhong et al., 2024; Hosseini et al., 2024). Further, following Rafailov et al. (2024), which showed that DPO can automatically learn a Q function, Qiu et al. (2024) devise a self-guided decoding algorithm limited for DPO models leveraging such property. However, despite these applications of adopting DPO models as off-the-shelf reward models or Q functions, none of existing work specifically targets improving such ability or investigating how to derive decent PRMs upon those off-the-shelf models.

## 7. Conclusion

We start with a theoretical proposition demonstrating that parameterizing the outcome reward as the log-likelihood ratios of the policy and reference models $\log \frac{\pi_\phi(y)}{\pi_{\text{ref}}(y)}$, a PRM can be intrinsically learned at the same time without any extra training requirements. We discuss its universality to instantiate different training objectives. In experiments, we demonstrate that various implicit reward modeling objectives outperform baselines on MATH, with substantially better trade-offs between accuracy and development overhead, particularly the CE loss. The performance of implicit PRMs can be further improved with majority voting. Further, scaling up instructions and responses benefit our implicit PRM, with the latter having a larger effect, but instructions should be relevant to downstream tasks while the diversity of responses does not bring gains. Surprisingly, training on extra Math-Shepherd step labels brings no further improvements to our implicit PRM trained on only outcome data.

## Impact Statement

This work aims to make process reward models (PRMs) generally more accessible by reducing training cost, thus promoting the advancement of inference time scaling and reinforcement learning. There may not be specific ethical consequences that are worthy of being particularly discussed here.

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

# A. Proof of Proposition

**Proposition A.1.** *Consider an ORM where the reward is parameterized by the log-likelihood ratio of two causal LMs, i.e.* $r_\phi(\mathbf{y}) := \beta \log \frac{\pi_\phi(\mathbf{y})}{\pi_{ref}(\mathbf{y})}$. *Define* $q_\phi^t(\mathbf{y}_{<t}, y_t) := \sum_{i=1}^t \beta \log \frac{\pi_\phi(y_i|\mathbf{y}_{<i})}{\pi_{ref}(y_i|\mathbf{y}_{<i})}$. $q_\phi^t$ *is the exponential average of* $r_\phi$ *at step* $t$.

$$q_\phi^t(\mathbf{y}_{<t}, y_t) = \beta \log \mathbb{E}_{\pi_{ref}(\mathbf{y}|\mathbf{y}_{\leq t})} e^{\frac{1}{\beta} r_\phi(\mathbf{y})} \tag{6}$$

*Proof.* The Proposition can be proven using mathematical induction.

Suppose response $\mathbf{y}$ has $T$ tokens.

**(1)** For $\forall t < T$, if $q_\phi^{t+1}(\mathbf{y}_{<t+1}, y_{t+1}) = \beta \log \mathbb{E}_{\pi_{ref}(\mathbf{y}|\mathbf{y}_{\leq t+1})} e^{\frac{1}{\beta} r_\phi(\mathbf{y})}$ holds, then $q_\phi^t(\mathbf{y}_{<t}, y_t) = \beta \log \mathbb{E}_{\pi_{ref}(\mathbf{y}|\mathbf{y}_{\leq t})} e^{\frac{1}{\beta} r_\phi(\mathbf{y})}$ would also hold.

**(2)** At $t = T$, $q_\phi^T(\mathbf{y}_{<T}, y_T) = r_\phi(\mathbf{y}) = \beta \log \mathbb{E}_{\pi_{ref}(\mathbf{y}|\mathbf{y}_{\leq T})} e^{\frac{1}{\beta} r_\phi(\mathbf{y})}$.

**proof of (1):**

$$
\begin{aligned}
\beta \log \mathbb{E}_{\pi_{ref}(\mathbf{y}|\mathbf{y}_{\leq t})} e^{\frac{1}{\beta} r_\phi(\mathbf{y})} &= \beta \log \mathbb{E}_{\pi_{ref}(\mathbf{y_{t+1}}|\mathbf{y}_{\leq t})} \mathbb{E}_{\pi_{ref}(\mathbf{y}|\mathbf{y}_{\leq t+1})} e^{\frac{1}{\beta} r_\phi(\mathbf{y})} \\
&= \beta \log \mathbb{E}_{\pi_{ref}(\mathbf{y_{t+1}}|\mathbf{y}_{\leq t})} e^{\frac{1}{\beta} q_\phi^{t+1}(\mathbf{y}_{<t+1}, y_{t+1})} \\
&= \beta \log \mathbb{E}_{\pi_{ref}(\mathbf{y_{t+1}}|\mathbf{y}_{\leq t})} \prod_{i=1}^{t+1} \frac{\pi_\phi(y_i|\mathbf{y}_{<i})}{\pi_{ref}(y_i|\mathbf{y}_{<i})} \\
&= \beta \log \prod_{i=1}^t \frac{\pi_\phi(y_i|\mathbf{y}_{<i})}{\pi_{ref}(y_i|\mathbf{y}_{<i})} \mathbb{E}_{\pi_{ref}(\mathbf{y_{t+1}}|\mathbf{y}_{\leq t})} \frac{\pi_\phi(y_{t+1}|\mathbf{y}_{\leq t})}{\pi_{ref}(y_{t+1}|\mathbf{y}_{\leq t})} \\
&= \beta \log \prod_{i=1}^t \frac{\pi_\phi(y_i|\mathbf{y}_{<i})}{\pi_{ref}(y_i|\mathbf{y}_{<i})} \sum_{y_{t+1}} \pi_{ref}(y_{t+1}|\mathbf{y}_{\leq t}) \frac{\pi_\phi(y_{t+1}|\mathbf{y}_{\leq t})}{\pi_{ref}(y_{t+1}|\mathbf{y}_{\leq t})} \\
&= \beta \log \prod_{i=1}^t \frac{\pi_\phi(y_i|\mathbf{y}_{<i})}{\pi_{ref}(y_i|\mathbf{y}_{<i})} \sum_{y_{t+1}} \pi_\phi(y_{t+1}|\mathbf{y}_{\leq t}) \\
&= \beta \log \prod_{i=1}^t \frac{\pi_\phi(y_i|\mathbf{y}_{<i})}{\pi_{ref}(y_i|\mathbf{y}_{<i})}
\end{aligned}
$$

**proof of (2):**

The conclusion is straightforward. Since $\pi$ is autoregressive, we have

$$r_\phi(\mathbf{y}) := \beta \log \frac{\pi_\phi(\mathbf{y})}{\pi_{ref}(\mathbf{y})} = \beta \log \prod_{i=1}^T \frac{\pi_\phi(y_i|\mathbf{y}_{<i})}{\pi_{ref}(y_i|\mathbf{y}_{<i})} = \sum_{i=1}^T \beta \log \frac{\pi_\phi(y_i|\mathbf{y}_{<i})}{\pi_{ref}(y_i|\mathbf{y}_{<i})}.$$

Since $\mathbf{y}_{\leq T} = \mathbf{y}$, the expectation $\mathbb{E}_{\pi_{ref}(\mathbf{y}|\mathbf{y}_{\leq T})}$ can be removed:

$$\beta \log \mathbb{E}_{\pi_{ref}(\mathbf{y}|\mathbf{y}_{\leq T})} e^{\frac{1}{\beta} r_\phi(\mathbf{y})} = \beta \log e^{\frac{1}{\beta} r_\phi(\mathbf{y})} = r_\phi(\mathbf{y}).$$

$\square$

# B. Frequently Asked Questions

## B.1. Why does the implicit reward not include a baseline $Z(X)$?

Our reward representation does not need a baseline, which differs from the implicit reward defined in Rafailov et al. (2023; 2024). The implicit reward in Rafailov et al. (2023; 2024) is derived from the entropy-regularized RL framework, and thus

a $Z(X)$ must be included to ensure the optimality of the trained policy model. However, this is not the case in our paper. Instead, we aim for a reward representation that enables a tractable Q value. We do not target an optimal policy in the entropy-regularized RL framework; Rather, our reward representation is defined and constructed from scratch, namely, any representation is acceptable as long as it gives a tractable way to estimate the Q value and makes Eq. (2) hold. Therefore, we do not need to follow the restrictions as Rafailov et al. (2023; 2024), and our reward representation does not necessarily relate to theirs.

Moreover, if a baseline term were added, one can prove that the following equation would not hold anymore, with the proof done by simply substituting the new reward representation (with a $Z(X)$) to the right-hand of the equation:

$$\sum_{i=1}^{t} \beta \log \frac{\pi_\phi(y_i \mid \mathbf{y} < i)}{\pi_{\text{ref}}(y_i \mid \mathbf{y} < i)} = \beta \log \mathbb{E}_{\pi_{\text{ref}}(\mathbf{y} \mid \mathbf{y} \leq t)} \left[ e^{\frac{1}{\beta} r_\phi(\mathbf{y})} \right]$$

### B.2. What are the advantages of CE loss compared to DPO?

As shown in Figure 5 and Figure 6, Implicit PRM with CE loss is more data efficient, while showing better performance when integrated with majority vote, presenting as an appealing alternative in practice as in many scenarios pair-wise data is hard to collect. Also, CE loss only requires one example for forwarding and backwarding which reduces memory overhead in RL training. Therefore, the generalization to unpaired losses remains valuable compared to pair-wise DPO in more data-constrained settings.

## C. Further Experiments

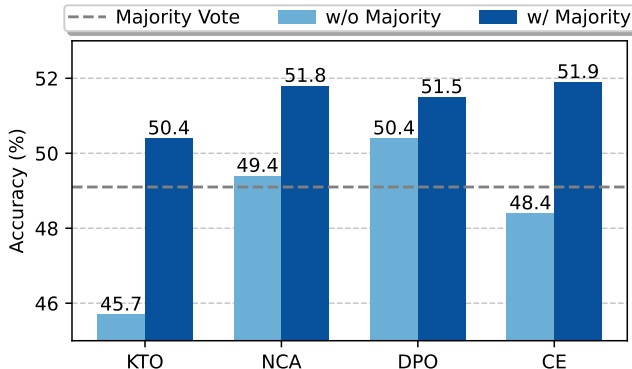

Figure 6: Results with majority voting. We present the averaged best-of-N accuracy across three testsets.

### C.1. Incorporating Majority Voting

Our Implicit PRMs can be integrated with majority voting to improve the performance even further. Previously, we apply our Implicit PRMs to score each response and pick the response with highest individual score as the final answer. However, when incorporating with majority voting, the scores of responses that lead to the same answer will be aggregated and the answer with the highest aggregated score will be selected as the final answer. We present the results averaged over different numbers of candidate solutions per problems across all three generated models in Figure 6.

We observe that our Implicit PRM can successfully adjust voting distributions, and achieves better results than using the Implicit PRM or majority voting separately. Particularly, KTO and CE variants gain the most from the integration, both of which fail to surpass majority voting alone but outperforms it through weighted best-of-N. It is also noteworthy that CE loss become the most effective when augmented with majority voting, once again demonstrating its potential.

### C.2. Are There Any Other Factors can Improve Implicit PRM Performance?

We consider potential factors that may influence the performance of Implicit PRMs, as listed below:

Table 5: Factors that may affect PRM performance. To our surprise, none of them consistently improve our implicit PRM.

| Setup | Mistral-7B-Inst-v0.2 | | | Llama-3.1-8B-Inst | | | Llama-3.1-70B-Inst | | | Avg. |
|---|---|---|---|---|---|---|---|---|---|---|
| | @4 | @16 | @64 | @4 | @16 | @64 | @4 | @16 | @64 | |
| Implicit PRM | 18.6 | 24.4 | 28.8 | 54.0 | 55.4 | 57.0 | 71.8 | 71.2 | 72.2 | 49.3 |
| + UltraFeedback | 19.4 | 24.4 | 29.0 | 53.8 | 55.0 | 55.8 | 71.6 | 70.6 | 72.2 | 49.2 |
| + UltraInteract (Code) | 19.2 | 24.6 | 28.0 | 54.6 | 54.0 | 56.8 | 71.4 | 70.8 | 70.0 | 49.2 |
| + Dedup. | 18.2 | 22.8 | 26.8 | 52.0 | 53.2 | 51.6 | 69.8 | 69.4 | 70.4 | 47.6 |
| + Base Resp. | 17.8 | 23.2 | 27.6 | 54.0 | 55.0 | 54.8 | 71.4 | 72.4 | 73.2 | 48.7 |
| + Step Label | 18.8 | 25.4 | 28.8 | 53.8 | 54.8 | 54.6 | 70.8 | 71.2 | 73.0 | 49.2 |

**Task-irrelevant Instructions**   We previously only consider math instructions. We now examine if increasing instructions diversity, even if the instructions are irrelevant to downstream tasks, can benefit Implicit PRMs. To this end, we incorporate general instructions from UltraFeedback (Cui et al., 2024) and coding instructions from UltraInteract (Yuan et al., 2024) into our training dataset. We directly use responses from the original datasets, but for UltraFeedback we only randomly select one pair for each instruction, instead of using all the pairs.

**Response Diversity**   We first conduct a deduplication on our preference dataset based on 8-gram overlap, aiming to verify if repeated responses hurt model performance. We then randomly replace four rollouts per instruction in the original training dataset with another four rollouts generated by Llama-3.1-8B-Base model.

**Training on Step Labels**   Our Implicit PRMs do not require step labels for training. However, we are interested in exploring whether augmenting them with step labels can further improve their performance. Based on the definition of process labels, we adjust the implicit reward of a step by increasing it for positive labels and decreasing it for negative ones. We use the labels obtained from our implemented Math-Shepherd, which has been demonstrated to be a strong implementation with step labels of high-quality (§4). We adapt KTO to a step-level version for optimization. Therefore, considering a $n$-step response with step labels $\{l^1, l^2, \ldots, l^n\}$, we conduct a *second stage* training on our current Implicit PRM to explicitly optimize the implicit reward: $\mathcal{L}_\phi = -\frac{1}{n} \sum_{t=1}^{n} \log \left( \sigma \left( l^t \cdot \left| r_\phi^t \right| \right) \right)$.

**Results**   We present results on Implicit PRM (DPO) in Table 5. In general, **none of these factors brings consistent gains.** (1) Both adding UltraFeedback and UltraInteract (code) instructions hurt the performance, with the former suffers more severely. This implies that training instructions deviating from the downstream task could undermine the performance of Implicit PRMs. (2) Regarding response diversity, we observe that the performance of deduplicating responses hurts the performance and is close to Implicit PRMs trained on similar amount of data. This indicates that repeated responses function similarly as others and are still beneficial before model performance saturates. Replacing part of original rollouts with those generated by the base model also fails to improve performance. (3) Conducting step-level KTO with extra process labels does not bring gains, reinforcing our claim that we can already train a strong PRM without process label. However, one should be cautious about concluding that stepwise labels are generally not helpful due to two factors in our experiments: Firstly, despite our efforts that lead to improved step annotation quality compared to previous work, the MCTS-based approach inevitably introduces noises in the data annotation process, as we discussed in §2; Secondly, our choice of algorithm may not be optimal. It is possible that more advanced PRM data annotation methods and training algorithms can finally integrate information from (noisy) stepwise labels into Implicit PRM.

