# OpenReview forum: "Free Process Rewards without Process Labels"
_ICML.cc/2025/Conference — ICML 2025 poster_

### Official Review · Reviewer_QUrp · 2025-03-11

**Overall Recommendation:** 3

**Summary:**

The paper introduces a method to train Process Reward Models (PRMs) without requiring expensive step-level annotations. By parameterizing outcome rewards as the log-likelihood ratio between a policy model and a reference model, PRMs can be implicitly derived from Outcome Reward Models (ORMs) trained on response-level data alone. This approach, validated on mathematical reasoning tasks, outperforms existing PRM methods (e.g., MCTS-based annotations) with significantly lower computational costs.

**Claims And Evidence:**

Yes

**Essential References Not Discussed:**

NA

**Experimental Designs Or Analyses:**

Yes

**Methods And Evaluation Criteria:**

Yes

**Other Comments Or Suggestions:**

NA

**Other Strengths And Weaknesses:**

Strengths:
1. Derives process rewards as log-likelihood ratios between policy and reference models, unifying preference learning (e.g., DPO) with PRM training.
2. Reduces training FLOPs compared to MCTS-based methods (e.g., Math-Shepherd), making PRMs accessible for resource-limited settings.
3. Compatible with diverse loss functions (DPO, CE, KTO, NCA), demonstrating flexibility beyond a single algorithm.


Weaknesses:
1. The core concepts and theoretical framework have already been established in [1]. Even with the proposed extensions, the theoretical contribution remains marginal.
2. While the authors suggest using cross-entropy (CE) loss to address scenarios with unpaired data, obtaining response-level labels may still pose challenges. Furthermore, the performance of CE loss is generally inferior to that of DPO.
3. Evaluated only on mathematical reasoning; generalizability to code generation or open-ended generation tasks is untested.


[1] Your Language Model is Secretly a Q-Function. https://arxiv.org/abs/2404.12358

**Questions For Authors:**

NA

**Relation To Broader Scientific Literature:**

NA

**Theoretical Claims:**

NA

---

> ### Author Rebuttal · Authors · 2025-04-01
>
> > 1. The core concepts and theoretical framework have already been established in [1]. Even with the proposed extensions, the theoretical contribution remains marginal.
>
> **A**: We note that the derivation of our work is different from [1] and provides a more general conclusion. [1] is tailored to DPO only and adopts a different reward representation (with a Z(X) baseline), with the implicit reward derived from entropy-regularized RL framework to ensure the optimality of the trained policy model. We aim for a reward representation that enables tractable Q value; we do not target and never claim an optimal policy in the entropy-regularized RL framework. Rather, our reward representation is **defined** and **constructed from scratch**, namely any representation is acceptable as long as it gives a tractable way to estimate Q value and makes Eq. (2) in line 137 hold. Compared to [1], this paper provides a fresh and more general perspective of implicit rewards, which we believe holds significant value.
>
> In this way, our paper provides a novel and more general theoretical framework which then leads to empirical benefits, e.g. the legibility of a CE loss, which offers an alternative when pairwise data is harder to collect than response-level labels, and scenarios that are more data-scarce. One may also explore other effective objectives beyond DPO and CE within our theoretical framework. Therefore, we believe our generalization to unpaired losses holds great theoretical and practical merits compared to previous works.
>
> [1] From r to Q∗: Your Language Model is Secretly a Q-Function. Rafailov et al. 2024.
>
> > 2. While the authors suggest using cross-entropy (CE) loss to address scenarios with unpaired data, obtaining response-level labels may still pose challenges. Furthermore, the performance of CE loss is generally inferior to that of DPO.
>
> **A**: We agree that in some cases response-level labels are still difficult to collect; our CE objective provides an alternative when such labels are available, and it can utilize the pairwise labels when they are easier to obtain.
>
> Also, CE loss has its own advantages over DPO. As shown in Figure 5 and Figure 6, Implicit PRM with CE loss is more data efficient, while showing better performance when integrated with majority vote, presenting as an appealing alternative in practice as in many scenarios pair-wise data is hard to collect. Also, CE loss only requires one example for forwarding and backwarding which reduces memory overhead in RL training. Therefore, the generalization to unpaired losses remains valuable compared to pair-wise DPO in more data-constrained settings.
>
> [2] Process Reinforcement through Implicit Rewards. Cui et al. 2025.
>
>
> > 3. Evaluated only on mathematical reasoning; generalizability to code generation or open-ended generation tasks is untested.
>
> **A:** Though we do not include other tasks in this paper due to our limited capacity and the limited space, after ICML submission deadline, there are recent works showing that Implicit PRM is helpful in best-of-N sampling on agent tasks [3], and adopting Implicit PRM for online RL, with brings substantial gains on coding [2] and function calling tasks [4].
>
> [3] AgentRM: Enhancing Agent Generalization with Reward Modeling. Xia et al. 2025.
>
> [4] Learning to Generate Structured Output with Schema Reinforcement Learning. Lu et al. 2025.

---

### Official Review · Reviewer_XrJW · 2025-03-19

**Overall Recommendation:** 3

**Summary:**

This paper proposes a new way of training process reward models without expensive fine-grained step-level annotations by training an ORM (reward modeled as log-likelihood ratios of the policy and the reference model) and using it as an implicit PRM. The authors show that their training of the implicit PRM is more data efficient than baseline PRM training strategies while also achieving good performance with Best-of-N sampling on the MATH dataset. Lastly, the paper contains analysis on how the their training strategy scales with different hyperparameters and loss functions for training the implicit PRM.

**Claims And Evidence:**

- Training implicit PRM is a more data-efficient approach than conventional PRM training recipes from prior work -- The authors show this successfully and convincingly
- Implicit PRM is performant and outperforms baselines: When it comes to these empirical results, the paper is lacking in the following ways:
    - [W1] *Limited Datasets*: All the results in this paper are based on the MATH dataset. However, a key advantage of any reward model is the ability to score generations from different distributions (even within math reasoning). The results would be more convincing if the best-of-N results were shown on several datasets such as GSM-8K, SVAMP, MMLU, AIME, etc.
   - [W2] *Is PRM better than ORM?*: In table 1, it appears that the Implicit PRM model is on average only a few points better than the other ORMs, so it is unclear why the training the implicit PRM is worthwhile (since training an ORM would be more efficient at inference time).

**Essential References Not Discussed:**

[1] https://arxiv.org/abs/2402.10963
[2] https://arxiv.org/abs/2403.04642

**Experimental Designs Or Analyses:**

See W1-3.

**Methods And Evaluation Criteria:**

In addition to W1 and W2, it is not clear if comparisons of Implicit PRM an other baselines is fair:

- [W3] *Fairness of Baseline Comparisons*: As pointed out in Table 2, at the 7B model scale, the Implicit PRM involves substantially more compute or GPU hours than the baseline, often by a factor comparable to 1-2x generations. In that case, when looking at the results in Table 1, it would be fair to compare the pass@4 performance of Implicit PRM with the pass@8 performance of baselines for Llama3 8B and pass@12 performance for Mistral. Based on the already slim margins by which their method beats baselines, most of the gains could be wiped away in this setting.

**Other Comments Or Suggestions:**

Found several typos in the main text: L 077 & L 384. Also check W1-3.

**Other Strengths And Weaknesses:**

Addressed in W1-3.

**Questions For Authors:**

- Given the PRMs provide step-level supervision, did the authors conduct any experiments to show their implicit PRM is more effective than other PRMs or ORMs at refinement [1] or RL training [2]?
- See the baselines or additional experiments requested in W1-3 above.

[1] https://arxiv.org/abs/2402.10963

[2] https://arxiv.org/abs/2403.04642

**Relation To Broader Scientific Literature:**

It proposes a new way of training an Implicit PRM without expensive fine-grained step-level annotations, which is more data efficient and performant than baseline PRM training strategies. They also make interesting connections to RL training of LLMs with implicit rewards as used in offline RLHF algorithms like DPO and show versatility in implicit PRM training objectives for paired and unpaired data.

**Theoretical Claims:**

Yes, in Appendix A.

---

> ### Author Rebuttal · Authors · 2025-04-01
>
> > [W1] Limited Datasets
>
> **A:** Though we do not include other tasks in this paper due to our limited capacity and the limited space, after ICML submission deadline, there are recent works showing that Implicit PRM is helpful in best-of-N sampling on agent tasks [1], and adopting Implicit PRM for online RL, with brings substantial gains on coding [2] and function calling tasks [3].
>
> [1] AgentRM: Enhancing Agent Generalization with Reward Modeling. Xia et al. 2025.
>
> [2] Process Reinforcement through Implicit Rewards. Cui et al. 2025.
>
> [3] Learning to Generate Structured Output with Schema Reinforcement Learning. Lu et al. 2025.
>
> Following reviewer’s suggestion, we also test on other math datasets and results are as follows. The policy model is Llama-3-70B-Instruct, and the baseline is Math-Shepherd in our implementations.
> | Method |  | AMC|(pass@1=36.1)  |   ||AIME|(pass@1=8.9)|
> |--------|:---:|:---:|:---|:---:|---:|:---:|:---|
> || @4|@16|@64||@4|@16|@64|
> | Math-Shepherd| 39.8|48.2|45.8| |17.8|18.9|17.8|
> | ImplicitPRM(DPO)|41.0|50.6|48.2||20.0|23.3|20.0|
>
> > [W2] Is PRM better than ORM?
>
> **A:** All ORMs in Table 1 are off-the-shelf from HuggingFace and are trained with different data and therefore not comparable to ours. They indeed consume much more data. For example, Eurus-RM-7B uses 287k pairs of data, SkyworkRM uses 465K, ArmoRM uses 1560K, while we only use 263K. Also, ORMs in Table 1 perform well on weak-to-strong settings which increases the average performance, but on Mistral-7B (20.4% vs 28.8% for best-of-64) and Llama-3.1-8B-Instruct (51.8% vs 57.0%), there are significant performance gaps to Implicit PRM.
>
> Moreover, Implicit PRM only needs to forward the response once, with the only difference being log probs are divided into steps and summed up respectively to find the minimum step reward. We acknowledge that Implicit PRM brings extra inference overhead due to the reference model and we have explored this issue in Appendix. As the cost mainly comes from the additional reference model, in Appendix B.3.1, we ablated it in both training and inference and presented results in Table 5. We found that in practice they can be removed at least in this paper’s setup, avoiding the extra compute while maintaining the performance. We also provided explanations in Appendix B.3.1 on why it may be removed. In such cases, Implicit PRM brings negligible overhead during inference compared to an ORM.
>
> > [W3] Fairness of Baseline Comparisons
>
> **A:** Please refer to the above response on how we can reduce the inference overhead of the reference model. Besides, following your suggestion, we present the compute-equivalent comparison between our best-of-4 and their best-of-8/12 as below:
> |Methods ||Mistral-7B-Inst-v0.2|| Llama-3.1-8B-Inst|
> |--------|:---:|:---:|:---|:---:|
> | |  @12|@48|@8|@32 |
> |Math-shepherd |22.2|25.2|51.4| 52.0 |
> ||@4|@16|@4|@16|
> |ImplicitPRM(DPO)|18.6| 24.4 |54.0| 55.4|
>
>
> > Given the PRMs provide step-level supervision, did the authors conduct any experiments to show their implicit PRM is more effective than other PRMs or ORMs at refinement [1] or RL training [2]?
>
> **A:** We supplemented comparison to ORMs with the same data as our Implicit PRM and still observed the superiority of Implicit PRM.
>
> ||||Mistral-7B-Inst-v0.2|||Llama-3.1-8B-Inst|||Llama-3.1-70B-Inst|Avg.|
> |---|---|---|---|---|---|---|---|---|---|---|
> ||@4|@16|@64|@4|@16|@64|@4|@16|@64||
> |ORM|18.6|23.8|26.4|50.8|50.2|52.8|68.8|71.0|69.4|48.0|
> |ImplicitPRM(DPO)|18.6|24.4|28.8|54.0|55.4|57.0|71.8|71.2|72.2|50.4|
>
> For RL training, a recent work [3] released after ICML submission deadline explored comprehensively how Implicit PRM can improve RL especially on sample efficiency compared to using verifiable outcome rewards only, namely the golden ORM. Results are shown as follows:
> |Method|AIME2024|AMC|MATH-500|MinervaMatch|OlympiadBench|LeetCode|LiveCodeBench|Avg.|
> |---|---|---|---|---|---|---|---|---|
> |GPT-4o|9.3|45.8|76.4|36.8|43.3|58.9|48.8|45.6|
> |Eurus-2-7B-SFT|3.3|30.1|66.2|32.7|29.8|21.7|17.8|28.8|
> |+RLw/GTOnly|20.0|47.0|73.2|36.4|35.4|28.3|26.7|36.9|
> |+RLw/GT+ImplicitPRM|26.7|57.8|79.2|38.6|42.1|33.3|28.6|43.9|
>
>
> [3] Process Reinforcement through Implicit Rewards. Cui et al. 2025.

---

> > ### Comment · Reviewer_XrJW · 2025-04-07
> >
> > I thank the authors for their effort in the response, but I have decided to keep my overall recommendation unchanged. My decision is primarily influenced by two factors: i) The compute-matched results are not very convincing since it shows the baseline outperforms implicit PRMs one model and trails on the rest -- so it is unclear what pattern holds for stronger base models like Qwen-Math; ii) The results of RL training with Implicit PRM are taken from another (concurrent) work unless the authors are arguing that the two methods are essentially the same implementationally, I am not convinced how it supports their work and would have liked to see a reproduction of this study in the author's setup.

---

> > > ### Author Response · Authors · 2025-04-09
> > >
> > > Thanks reviewer for the engagement! To address the concerns, we added two experiments:
> > >
> > > 1. More rigorous compute-match experiments. We previously presented best-of-4 of Implicit PRM versus best-of-8 of baseline on Llama-8B, and best-of-12 of baseline on Mistral-7B, following reviewer’s suggestion. However, according to Table 2, the GPU time costs of Implicit PRM in real-world practice is 301.6/200.9=1.5 times of baseline’s cost on Mistral-7B, 241.7/171.1=1.41 times on Llama-8B, and 122.2/111.1=1.1 on Llama-70B. Hence, the fair comparison should be **best-of-4 of Implicit PRM vs. best-of-6 of baselines on Mistral and Llama-8B**.
> > >
> > > Moreover, as indicated in Table 5 in Appendix B.3.1, we can remove the reference model at inference to reduce overhead in some cases, reaching the same level of inference efficiency as our baselines. We evaluated the reference-free version of Implicit PRM under the same budget as our baselines.
> > > Results are as follows, from which we can see Implicit PRM, with or without a reference model, is comparable on Mistral-7B and outperforms Math-Shepherd by a large margin on Llama-8B.
> > > | Method | Mistral |  | Llama |  |
> > > |---|:---:|:---:|:---:|:---:|
> > > |  | @6 | @24 | @6 | @24 |
> > > | Math-Shepherd (Trained on same data) | 20 | 24.6 | 49.8 | 51.6 |
> > > | Implicit PRM (DPO, w/o ref) | 18.8 | 24.8 | 54.6 | 55.2 |
> > > |  | @4 | @16 | @4 | @16 |
> > > | Implicit PRM (DPO) | 18.6 | 24.4 | 54 | 55.4 |
> > >
> > > 2. Following reviewer’s suggestion, we implemented RLVR [1] with PPO, using only ground-truth outcome rewards, and then integrated it with online updated Implicit PRM, where the rollouts assessed by golden outcome rewards can be used to train Implicit PRM as in Eq. (5). We used publicly available instructions on [Huggingface](https://huggingface.co/datasets/PRIME-RL/Eurus-2-RL-Data). We ran both for 160 steps within the discussion period and tested on coding and math benchmarks. Results show that Implicit PRM augmented PPO is generally better than GT-only PPO across various benchmarks.
> > > |  | HumanEval | MBPP | LeetCode | MATH500 | AMC | AIME | Minerva | OlympicBench | avg |
> > > |---|:---:|:---:|:---:|:---:|:---:|:---:|:---:|:---:|:---:|
> > > | PPO w/ GT Only | 72.0 | 56.1 | 28.3 | 71.8 | 48.2 | 6.7 | 35.3 | 34.5 | 44.1 |
> > > | PPO w/ GT and Implicit PRM | 72.0 | 59.1 | 27.8 | 76.4 | 48.2 | 20.0 | 39.7 | 37.9 | 47.6 |
> > >
> > > [1]  Tulu 3: Pushing Frontiers in Open Language Model Post-Training. Lambert et al. 2024.

---

### Official Review · Reviewer_JdZn · 2025-03-19

**Overall Recommendation:** 3

**Summary:**

This paper shows that the PRM can be obtained implicitly without additional training by parametarization.

**Claims And Evidence:**

From the table, the gain delta from the proposed PRM is not that large, and on Mistra-7B is not helping, Besides, the pass@1 is not clear if it helps, would be great to show a curve here.

**Essential References Not Discussed:**

Looks comprehensive pile of relevant work.

**Experimental Designs Or Analyses:**

Looks reasonable to me, with the ORM and PRM as baselines, and the sufficient base model as well as Reward model to show the ablations and gains.

**Methods And Evaluation Criteria:**

Looks good.

**Other Comments Or Suggestions:**

Would like to know more about the pass@k from 1 to 16, how the curve looks like, particularly for pass@1, how much it helps. Besides how many runs for this metric, would be great to show mean/std here.

**Other Strengths And Weaknesses:**

The paper is in general a good shape, that not an expert in this domain can still follow the flow easily. Both theoretical and empirical results looks good to me.

**Questions For Authors:**

As in comments and above. The main concern is the empirical gain, if it's really showing the consistent gain. Would be great if this part can be explained more (particularly, why Mistral-7B no gain, and pass@4 for Llama/pass@64 for Llama-70B

**Relation To Broader Scientific Literature:**

If this generalize to many other domain, would be a big gain for efficient RL scaling for LLM

**Theoretical Claims:**

Looks good, but not an expert to verify the correctness.

---

> ### Author Rebuttal · Authors · 2025-04-01
>
> > From the table, the gain delta from the proposed PRM is not that large, and on Mistra-7B is not helping, Besides, the pass@1 is not clear if it helps, would be great to show a curve here.
>
> **A:** This might be  a misinterpretation of Table 1. In fact, our approach achieves very strong performance.
>
>  We should compare the best-of-N performance to pass@1(greedy decoding w/o using reward models) to evaluate the effectiveness of the reward models. The pass@1, i.e. best-of-1, of each generation model is highlighted in the header and the caption. The first column for each model denotes the best-of-4 accuracy and is already much higher than pass@1 across the board, confirming the strong performance of our approach. Rows in the “open-source reward models” block  are trained with different and much larger and more engineered data, and therefore are not comparable to ours. Those under “our implementations” are reimplementations of previous works controlling for confounding factors such as the base model and data, for fair comparisons.
>
> Contrary to the reviewer’s observation that “on Mistral-7B it is not helping,” the gains on Mistral-7B are actually the largest:  the pass@1 is 9.6% while best-of-64 increased to 28.8% for our DPO model, a 19.2 absolute gain. Improvements are still observed even when we use the 8B-sized PRM to assist 70B-sized generation models, namely in the weak-to-strong setup, evidenced by best-of-64 of 72.2% compared to pass@1 of 63.2%. Importantly, our reported performance improvement on MATH is challenging to achieve: contextualizing it within recent impactful literature [1,2], a <3% improvement on best-of-64 and 2% improvement compared to baselines is usually considered substantial as a significant improvement. We think our Implicit PRMs achieve strong performance in our experiments.
>
> We also note that why we did not surpass the off-the-shelf baseline RLHFlow-8B-Mistral-Data can be explained by the unfair comparison, due to their advantages from training models using on-policy rollouts. Our evidences include: (1) RLHFlow-8B-Mistral-Data uses **on-policy** rollouts from Mistral-7B while ours use off-policy rollouts from Llama-3.1-8B-Instruct; (2) The same approach with rollouts from DeepSeek models underperforms ours; (3) Our approach outperforms RLHFlow-8B-Mistral-Data on Llama-3.1-8B-Instruct by a large margin.
>
> Following the reviewer's suggestion, we plot the curve of pass@N and bes-of-N in this [figures](https://ibb.co/Qvt4GQgM) anonymously. from the figure we can see that our method consistently outperforms baselines when increasing the number of candidate responses (N).
>
> [1] Math-shepherd: Verify and reinforce llms step-by-step without human annotations. Wang et al. 2023.
>
> [2] Improve mathematical reasoning in language models by automated process supervision. Luo et al. 2024.
>
> > Whether this generalize to many other domain.
>
> **A:** Though we do not include other tasks in this paper due to our limited capacity and the limited space, after ICML submission deadline, there are recent works showing that Implicit PRM is helpfull in best-of-N sampling on agent tasks [3], and adopting Implicit PRM for online RL, with brings substantial gains on coding [4] and function calling tasks [5].
>
> [3] AgentRM: Enhancing Agent Generalization with Reward Modeling. Xia et al. 2025.
>
> [4] Process Reinforcement through Implicit Rewards. Cui et al. 2025.
>
> [5] Learning to Generate Structured Output with Schema Reinforcement Learning. Lu et al. 2025.
>
> > Would like to know more about the pass@k from 1 to 16, how the curve looks like, particularly for pass@1, how much it helps. Besides how many runs for this metric, would be great to show mean/std here.
>
> **A:** Thanks for the suggestion. The curve of pass@N and bes-of-N can be found [here](https://ibb.co/Qvt4GQgM).
> For each experiment we only run for one time since our early experiments show that there is little variance across runs and thus this has been a standard practice mainly due to the cost of the experiments [1,2,6].
>
> [6] Autopsv: Automated process-supervised verifier. Lu et al. 2024.
>
> > The main concern is the empirical gain, if it's really showing the consistent gain. Would be great if this part can be explained more (particularly, why Mistral-7B no gain, and pass@4 for Llama/pass@64 for Llama-70B
>
> **A:** Please see above responses on empirical gains. Particularly, our method has achieved substantial gains (pass@1: 9.6% -> best-of-64: 28.8%) on Mistral-7B rather than no gains, and we only underperform the baseline trained on on-policy Mistral-7B rollouts. Regarding pass@4 for Llama/pass@64 for Llama-70B (actually they are best-of-4 and best-of-64), though we did not achieve the highest accuracy, our performance is very close to the best-performing baseline, only with a difference of 0.4% and 0.8% respectively. We kindly ask the reviewer reevaluate our submission after these clarifications on the empirical gains.

---

> > ### Comment · Reviewer_JdZn · 2025-04-02
> >
> > Thanks for addressing my concerns! Raised my rating.

---

### Official Review · Reviewer_kaar · 2025-03-23

**Overall Recommendation:** 3

**Summary:**

This paper introduces a method to create a process reward model (PRM) without the need for expensive step-by-step annotations. The authors propose that an implicit PRM can be derived by training an ORM using only response-level labels, by parameterizing the outcome reward as a log-likelihood ratio between the policy and reference models. Their experiments on MATH show that this implicit PRM performs better than a strong MCTS-based baseline (Math-Shepherd) while using significantly less training data. The model’s performance further improves with majority voting, and scaling up instructions and responses boosts effectiveness, with responses contributing more to improvements.

## update after rebuttal
I have read the author responses, and my evaluation remains the same. I feel the proposed dense rewards need to be tested as dense rewards for online RL, or at lease beam search at test-time (common in recent literature as I mention below), to know if they are empirically truly effective in improving search efficiency.

**Claims And Evidence:**

The paper claims that 10-30x reduction in the overhead needed to train PRMs, if we instead just train ORMs with re-parameterized rewards as in DPO, and then use the proposed prefix level scores to evaluate intermediate steps. The experiments on Math benchmarks support this claim.

**Essential References Not Discussed:**

Improve Mathematical Reasoning in Language Models by Automated Process Supervision Luo et. al.

**Experimental Designs Or Analyses:**

Yes, the models and benchmarks chosen are valid and sound.

**Methods And Evaluation Criteria:**

Yes, the evaluation criteria for BoN follows Lightman et. al.. But more recent works evaluate PRMs with beam search or as dense rewards in RL (see comments below). The benchmarks and models chosen are indeed standard.

**Other Comments Or Suggestions:**

- L77 typo, parameterizin ---> parameterizing

**Other Strengths And Weaknesses:**

Strengths
- The implicit PRM does not need training or data collection beyond what is needed for an ORM.
- The analysis in 5.1 seems to suggest that the trained PRMs are indeed data efficient and performance improves as training data is scaled consistently.


Weaknesses
- The paper only evaluates the PRM as an ORM, to re-rank responses during BoN. The true test of PRM would be to run beam search as in Snell et. al. or to use it as dense rewards for online RL, as on Setlur et. al.
- The gains are small on poorer models like Mistral-7B, and in general, the gains are around 5% when averaged across more performant models on MATH. So, it seems that the PRM is not very useful on hard questions.

**Questions For Authors:**

- Can the authors provide some discussion on how to optimally scale instructions and responses when collecting the training data for training PRMs? Also, for this, are the samples sampled IID from the base model or balanced across correct and incorrect samples?
- Does the trained PRM extrapolate, i.e., how well does a PRM train on Mistral7B transfer to LLama and vice-versa, or for the same base model how does it extrapolate from GSM8k --> MATH or vice-versa?
- Do you think that this has would also work on domains other than MATH, like PRMs for instruction tuning, or coding?

**Relation To Broader Scientific Literature:**

The main benefit of the approach proposed in this paper is that the it only requires training an ORM with implicit rewards in order to turn it into a PRM that can score partial generations. In contrast, other prior works like MathShephard, OmegaPRM and PAVs, train PRMs to predict value functions, which requires a larger collection of data, which roughly scales linearly with the sequence length. Though, this paper does not evaluate the trained PRM during beam search or for RL, which is the main way Snell et. al. and Setlur et. al. evaluated and used PRMs to scale inference time compute.

**Theoretical Claims:**

Yes, I checked the validity of Proposition 3.1 in Appendix A.

---

> ### Author Rebuttal · Authors · 2025-04-01
>
> We thank the reviewer for the constructive comments. Here are our responses.
>
> > The true test of PRM would be to run beam search or to use it as dense rewards for online RL.
>
> **A:** We choose best-of-N as our setup because it is a standard practice in recent literature and presents as a valuable approach for inference-time scaling [1,2,3]. Testing PRMs for online RL is of course valuable, but can introduce additional confounding factors and much more overhead. This paper aims to isolate the quality of the reward model, so best-of-N sampling provides relevant and convincing evidence for our claim.
>
> That being said, we notice that a recent work [4] after the ICML ddl closely resembles the reviewer's suggestion, which applies our implicit PRM in online RL and achieved strong performance and sample efficiency across various benchmarks. We think this proves our approach's potential in online RL.
>
> We are now running the beam search experiments that the reviewer suggested. They are slow and we will provide update as soon as possible.
>
> [1] Let’s Verify Step by Step. Lightman et al. 2023.
>
> [2] Math-shepherd: Verify and reinforce llms step-by-step without human annotations. Wang et al. 2023.
>
> [3] Scaling LLM Test-Time Compute Optimally can be More Effective than Scaling Model Parameters. Snell et al. 2024.
>
> [4] Process Reinforcement through Implicit Rewards. Cui et al. 2025.
>
> > The gains are small on poorer models like Mistral-7B... It seems that the PRM is not very useful on hard questions.
>
> **A:** This might be  a misinterpretation of Table 1.  In fact, our approach achieves very strong performance on these questions.
>
> We note that the pass@1 performance of each generation model is highlighted in the header and the caption, and the first column for each model denotes the best-of-4 accuracy and is already much higher than pass@1. We should compare the best-of-N performance to pass@1 to see the absolute gain of each model.
>
> Contrary to the reviewer’s observation, the gains on Mistral-7B are actually the largest, with the pass@1 being 9.6% while best-of-64 increased to 28.8% for our DPO model. Compared to our implemented Math-Shepherd and AutoPSV, we achieved significant average improvements of nearly 3% and 5%.
>
> As for the reviewer’s second point: a 5% average improvement on the MATH benchmark is challenging to achieve: contextualizing it within recent impactful literature [2, 5], a <3% improvement on best-of-64 and 2% improvement compared to baselines is usually considered substantial. Therefore, the average performance gains of our implicit PRM demonstrate its effectiveness on these challenging questions.
>
> We also add comparisons to ORMs below; both are trained with the same data. The trend is consistent with that in the paper and we observe strong performance of our Implicit PRM.
> ||||Mistral-7B-Inst-v0.2|||Llama-3.1-8B-Inst|||Llama-3.1-70B-Inst|Avg.|
> |---|---|---|---|---|---|---|---|---|---|---|
> ||@4|@16|@64|@4|@16|@64|@4|@16|@64||
> |ORM|18.6|23.8|26.4|50.8|50.2|52.8|68.8|71.0|69.4|48.0|
> |ImplicitPRM(DPO)|18.6|24.4|28.8|54.0|55.4|57.0|71.8|71.2|72.2|50.4|
>
> [5] Improve mathematical reasoning in language models by automated process supervision. Luo et al. 2024.
>
> > How to optimally scale instructions and responses
>
> **A:** According to our experiments in section 5.1, scaling up instructions from the same tasks to downstream tests and scaling up responses are both is helpful. We did not test data balance for the DPO objective, but we did observe benefits of that for CE loss.
>
> > Does the trained PRM extrapolate
>
> **A:** Yes, Implicit PRM is able to transfer across model families and tests of the same task. For model transfer, all Implicit PRMs in Table 1 are trained from Llama-3.1-8B-Inst with its on-policy rollouts, and existing results have shown their effectiveness on improving Mistral-7B and Llama-3.1-70B-Inst models. For task transfer, we add an experiment to train our model solely on GSM8K and test on MATH as follows, and surprisingly, results show that the overall performance remains comparable to that trained on all instructions, indicating the superior task transferability of Implicit PRM.
>
> ||||Mistral-7B-Inst-v0.2|||Llama-3.1-8B-Inst|||Llama-3.1-70B-Inst|Avg.|
> |---|---|---|---|---|---|---|---|---|---|---|
> ||@4|@16|@64|@4|@16|@64|@4|@16|@64||
> |ImplicitPRM(DPO)|18.6|24.4|28.8|54.0|55.4|57.0|71.8|71.2|72.2|50.4|
> |ImplicitPRM(DPO on GSM8k)|18.6|24.8|28.2|54.6|54.8|57.0|71.0|71.0|72.6|50.3|
>
> > If Implicit PRM works on other domains
>
> **A:** Though we do not include other tasks in this paper due to our limited capacity and the limited space, after ICML ddl, there are recent works showing that Implicit PRM is helpful in best-of-N sampling on agent tasks [6] and online RL, with substantial gains on coding [4] and function calling tasks [7].
>
> [6] AgentRM: Enhancing Agent Generalization with Reward Modeling. Xia et al. 2025.
>
> [7] Learning to Generate Structured Output with Schema Reinforcement Learning. Lu et al. 2025.

---

### Official Review · Reviewer_nJ5A · 2025-03-26

**Overall Recommendation:** 3

**Summary:**

Verifiers, such as process reward PRMs and ORMs, evaluate LLMs' partial or full responses, providing feedback and pushing the boundaries of LLMs' ability to solve complex reasoning tasks. . PRMs provide better, fine-grained feedback than ORM based on the nature of their training procedure. However, training PRMs is more challenging than training ORMs because it requires annotating every intermediate step. This paper argues that a strong PRM can be derived at no additional cost from training an ORM. The observation is that using the closed-form solution of the reward model to learn a reward and policy jointly, similar to the DPO method, a PRM can be automatically learned during training. The paper extends this idea beyond DPO-style algorithms by incorporating cross-entropy (non-binary objectives). The paper also experiments with three different LLMs, along with several variants of optimization objectives based on DPO, demonstrating the robustness of this approach. The paper also includes several ablation studies highlighting the importance of increasing the number of responses per instruction and scaling the number of instructions.

**Claims And Evidence:**

Below are the claims and evidence presented in the paper:

- The paper claims that PRM performs better than ORM. They support this claim by referencing other studies in the literature that demonstrate this advantage and conducting experiments to compare both PRM and ORM.

- Another claim is that PRM can be easily learned by optimizing a DPO-style objective, which leverages the closed-form solution of the reward model. The paper provides empirical evidence showing that their implicit PRM performs well in practice compared to ORM and is competitive with traditionally trained PRM.

**Essential References Not Discussed:**

None

**Experimental Designs Or Analyses:**

The experimental design and analyses are sound. The paper proposes to study their idea across several models to demonstrate its generality. Additionally, the paper provides several key ablation studies to identify where the performance gains are coming from in the proposed approach.

**Methods And Evaluation Criteria:**

Yes, the proposed method and evaluation criteria are suitable for the problem and application at hand. The authors aim to train the PRM more efficiently to address downstream tasks that require complex reasoning, such as math instruction datasets and chat instruction-following tasks.

**Other Comments Or Suggestions:**

None

**Other Strengths And Weaknesses:**

N/A

**Questions For Authors:**

Below is list of questions that I have:
- DPO optimizes pairs to eliminate the intractable computation of the normalization constant. How do you manage the normalization constant in the cross-entropy (CE) update? If you are making assumptions about the normalization constant, what assumption are you making, and why is it feasible?

- The paper concludes that the implicit DPO reward outperforms the other objectives studied. However, implicit DPO was originally proposed in [1], where it was used to learn a PRM and apply it during inference. Given this context, it is unclear what novelty this paper offers, as the best algorithm was introduced in previous work, and the single-sample algorithm has issues with the normalization constant.

- In Figure 4, why does increasing the number of scaling instructions hurt performance?



[1] Treebon: Enhancing inference-time alignment with speculative tree-search and best-of-n sampling Qui et al. 2024

**Relation To Broader Scientific Literature:**

The key contribution of the paper relates to the broader scientific literature by focusing on improving the efficiency of training Probabilistic Relational Models (PRM). The authors observe that PRMs outperform Ordinary Relational Models (ORMs), but the drawback is that they are costly to train. Their proposed solution aims to reduce the complexity of PRMs, enabling large language models (LLMs) to learn more efficiently since PRMs provide more informative data.

**Theoretical Claims:**

I've reviewed the theoretical claims, and the proof appears to have some issues, but I could be wrong. The authors assert that $E_x[f(x) g] = E_x[f(x)] g$, where $g = E_{y<t+1}$ and $f(x) = E_{y_t < y_t}$. This assertion is problematic because $g$ is not a constant. Additionally, the second step of proof (1) seems to have some flaws as well. Furthermore, the derivation of cross-entropy loss is missing the normalization constant Z(x). The reason for DPO to minimize the pairwise loss is to cancel the intractable normalization constant.

---

> ### Author Rebuttal · Authors · 2025-04-01
>
> We thank the reviewer for the valuable feedback and are glad that you find the method is suitable for the problem and the experiments and analysis are sound. Here are our responses.
>
> > the proof appears to have some issues. Furthermore, the derivation of cross-entropy loss is missing the normalization constant Z(x).
>
> **A**: Thanks for the careful reading, but we believe there are misunderstandings.
>
> As the reviewer uses different notations and does not mention the line of proof, we would like to kindly ask the reviewer for clarification on the specific steps in the proof, so that we can better address the concern.
>
> Regarding the baseline $Z(X)$ in the cross entropy objective: Please note that our reward representation does not need a baseline $Z(X)$, which is different from the implicit reward defined in [2,3]. The implicit reward in [2,3] is derived from the entropy-regularized RL framework, and a $Z(X)$ must be included to ensure the optimality of the trained policy model. However, this is not the case in our paper. Instead, we aim for a reward representation that enables tractable Q value; we do not target and never claim an optimal policy in the entropy-regularized RL framework. Rather, our reward representation is **defined** and **constructed from scratch**, namely any representation is acceptable as long as it gives a tractable way to estimate Q value and makes Eq. (2) in line 137 hold. Therefore, we do not need to follow the restrictions as [2,3] and our reward representation does not necessarily relate to theirs.
>
> Moreover, if a baseline term $Z(X)$ were added, one can  prove that **the following equation would not hold** anymore, with the proof done by simply substituting the new reward representation (with a $Z(X)$) to the right-hand of the equation:
>
> $$
>
> \sum_{i=1}^{t} \beta \log \frac{\pi_\phi(y_i \mid \mathbf{y}_{<i})}{\pi_\text{ref}(y_i \mid \mathbf{y}_{<i})} = \beta \log \mathbb{E}_{\pi_\text{ref}(\mathbf{y} \mid \mathbf{y}_{\leq t})} \left[ e^{\frac{1}{\beta} r_\phi(\mathbf{y})} \right]
>
> $$
>
> I.e., proposition 3.1 will not consistently hold for all ORM objectives if we follow the reward representation as [2,3], and we won’t be able to find a tractable Q value.
>
> We plan to add this discussion in the next version.
>
> > How do you manage the normalization constant in the cross-entropy (CE) update?
>
> **A**: Please refer to the response on $Z(X)$ above. As stated in Proposition 3.1, we do not need a baseline in our reward representation.
>
> > The paper concludes that the implicit DPO reward outperforms the other objectives studied. Implicit DPO was originally proposed in [1]. Given this context, it is unclear what novelty this paper offers, as the best algorithm was introduced in previous work.
>
> **A**: First, [1] directly adopts the theory from [3] which only applies to DPO. However, as discussed above, our work also provides a more general proposition with solid theoretical contribution, applying to any ORM objectives including CE loss as long as they use our reward representation, compared to both [1] and [3] that are tailored to DPO. Compared to these previous works, this paper provides a fresh and more general perspective of implicit rewards, which we believe holds significant value.
> Second, as shown in Figure 5 and Figure 6, Implicit PRM with CE loss is more data efficient than its DPO counterpart while showing better performance when integrated with majority vote, presenting as an appealing alternative in practice as in many scenarios pair-wise data is hard to collect. Also, the CE variant  requires only one example for forwarding and backwarding while the DPO variant has to consider a pair of  examples at the same time. As a result, the CE variant reduces memory overhead in RL training, as observed by a recent work that directly adopts our method (published after the ICML submission deadline) [3]. Therefore, we believe our generalization to unpaired losses holds great theoretical and practical merits compared to previous works.
>
> > In Figure 4, why does increasing the number of scaling instructions hurt performance?
>
> **A:**  We’d like to clarify: for most cases, increasing the number of instructions improves model performance. The only exception is using all instructions to train 8B-sized PRMs and test on Llama-3.1-70B-Inst. As the increased number of instructions come with Llama-3.1-8B-Instruct generated responses, we conjecture that the performance drop can be attributed to PRMs overfitting to responses from the small model and being unable to generalize to larger models. This discussion will be added in the revision.
>
> [1] Treebon: Enhancing inference-time alignment with speculative tree-search and best-of-n sampling Qui et al. 2024
>
> [2] Direct Preference Optimization: Your Language Model is Secretly a Reward Model. Rafailov et al. 2023.
>
> [3] From r to Q∗: Your Language Model is Secretly a Q-Function. Rafailov et al. 2024.

---

### Decision · Program_Chairs · 2025-05-01

**Decision:**

Accept (poster)

**Comment:**

This paper presents a novel method for training implicit Process Reward Models (PRMs) using only outcome labels, without relying on step-by-step process supervision. The approach is inspired by DPO, where the reward function is parameterized as the log-likelihood ratio between the trained and the reference model.
The proposed method offers two main advantages: (1) it achieves better performance compared to Outcome Reward Models (ORMs) on MATH, and (2) it is more data efficient than prior PRM training approaches. These claims are supported by well-designed experiments, and all reviewers express a favorable opinion of the work.

One minor concern, raised by reviewers kaar, XrJW, and QUrp, is that the application of the implicit PRM is currently limited to Best-of-N sampling and has only been evaluated on math datasets. It would be a valuable addition to provide a thoughtful analysis of potential broader applications (e.g., RL training).

Overall, this work introduces a compelling approach to training implicit PRMs without stepwise supervision. I recommend acceptance.